# Functional Analysis of Oligoadenylate Synthetase in the Emu (*Dromaius novaehollandiae*)

**DOI:** 10.3390/ani14111579

**Published:** 2024-05-27

**Authors:** Keisuke Sato, Teppei Nakamura, Masami Morimatsu, Takashi Agui

**Affiliations:** Laboratory of Laboratory Animal Science and Medicine, Faculty of Veterinary Medicine, Hokkaido University, Sapporo 060-0818, Japan; nakamurate@vetmed.hokudai.ac.jp (T.N.); mmorimat@vetmed.hokudai.ac.jp (M.M.); agui@vetmed.hokudai.ac.jp (T.A.)

**Keywords:** 2′-5′ oligoadenylate synthetase, emu, West Nile virus

## Abstract

**Simple Summary:**

Oligoadenylate synthetase (OAS) is a conserved antiviral protein found in several animal species. Among birds, only the ostrich has two genes, *OAS1* and *OASL*, which show oligoadenylate synthetic activity and inhibition of flavivirus-specific genome replication, respectively. However, it remains unclear whether OAS duplication with the separated function is unique to the ostrich. Therefore, we examined the OASs of the emu, which are closely related to the ostrich. We sequenced and cloned emu OAS genes and analyzed their functions. The results showed that the two emu OAS genes had amino acid sequence homologies of 80% for OAS1 and 78% for OASL compared to those of ostriches. The amino acid sequences related to the enzymatic function were almost identical to those of ostriches. Emu OAS1 only showed OAS activity, whereas emu OASL only inhibited flaviviral replication. These results indicate that emus have characteristics similar to ostrich in terms of OAS genes. This study provides insights into the evolution of viral defense by OAS protein family in Palaeognathae.

**Abstract:**

2′-5′-oligoadenylate synthetase (OAS) is one of the proteins that act as a defense mechanism against foreign RNA in cells. OAS has two functions: an antiviral effect against a wide range of virus species via the OAS/RNase L pathway with synthesized oligoadenylates and inhibition of viral replication specific to viruses of the genus *Flavivirus*, which is independent of enzymatic activity. Several birds have been reported to possess only one type of OAS family member, OASL, which has both enzymatic activity and inhibitory effects on flaviviral replication. However, the ostrich has two types of OASs, *OAS1* and *OASL*, which show different functions—enzymatic and anti-flaviviral activities, respectively. In this study, emu OASs were cloned to investigate their sequence and function and elucidate the role of OASs in emus. The cloning results showed that emus had OAS1 and OASL, suggesting that emu OASs were more closely related to ostrich than to other birds. Functional investigations showed that emu OAS1 and OASL had enzymatic and anti-flaviviral activities, respectively, similar to those of the ostrich. Emus and ostriches are evolutionarily different from most birds and may be more closely related to mammalian OAS diversity.

## 1. Introduction

Innate immunity is important for the maintenance of homeostasis in multicellular organisms. Pathogenic microorganisms threaten host homeostasis, and hosts have developed various mechanisms to counteract these pathogens. Interferons (IFNs) have been developed in vertebrate cells to induce proteins that effectively eliminate viruses that invade cells [1,2,3]. IFNs are induced by various stimuli—including melanoma differentiation-associated gene 5 (MDA5), retinoic acid-inducible gene I (RIG-I), and viral RNA-sensing proteins—that are expressed in virus-infected cells [4,5,6]. Genes induced by IFNs are referred to as interferon-stimulated genes (ISGs), and several hundreds of such genes have been identified.

2′-5′-oligoadenylate synthetase (OAS) is a protein that acts as a defense mechanism against foreign RNA in cells. OAS is one of the ISGs that has been discovered as an enzyme that synthesizes oligoadenylates [7,8]. OASs recognize double-stranded RNA of specific lengths and synthesize oligoadenylates from intracellular adenosine triphosphate (ATP). Oligoadenylates dimerize RNase L into its active form, which recognizes and fragments RNA, resulting in decreased cellular protein expression. The fragmented RNA triggers RNA-sensing machinery, such as MDA5 and RIG-I, in the cell and enhances IFN expression. This activates the entire cellular antiviral machinery and promotes the expression of OAS as an ISG [9,10]. In this mechanism, OASs play a dual role as an enhancer and a sensor of the antiviral machinery. In humans, genome-wide association studies have shown that single nucleotide polymorphisms in OASs can alter the severity of infection and have a significant impact on virus elimination in cells [11].

The OAS family is a widely conserved protein family which has been reported in many vertebrates and sponges [12,13,14,15]. There are several reports on OASs in birds, such as chickens (*Gallus gallus*), ducks (*Anas platyrhynchos*), geese (*Anser cygnoides*), and ostriches (*Struthio camelus australis*) [16,17,18,19,20]. Among these, chickens, ducks, and geese have one OAS gene, 2′-5′-oligoadenylate synthetase-like (*OASL*), which has both enzymatic activity and inhibitory effects on flavivirus replication. In contrast, ostriches have been shown to have two OAS genes, *OAS1* and *OASL*, which have enzymatic activity and inhibitory effects on flavivirus replication, respectively [16,18,19].

In addition, mouse Oas1b has been reported to inhibit viral genome replication, specifically in viruses of the genus *Flavivirus* [21,22]. This activity has been suggested to be independent of the RNase L pathway because replication inhibition is still observed when genes constituting the OAS/RNase L pathway are knocked out; however, the details of their functions have not been clarified. Mice have 12 OAS families, but only Oas1b has flavivirus replication inhibitory activity. Two of the mouse OASs have been reported to have enzymatic activity, but the other nine OASs have neither enzymatic activity nor flaviviral replication inhibition, and their roles remain unknown [23].

Extant birds are classified into two groups: Palaeognathae and Neognathae. Neognathae consists of Galloanserae (chickens, ducks, and geese) and Neoaves. Recent studies using numerous genes support that the ostrich was the first to diverge among the palaeognaths, and that there is a gap between the ostrich and other palaeognaths. The emu is a flightless bird classified as ratite, a group of birds also including kiwi and rheas. The emu is closely related to the ostrich among palaeognaths in terms of molecular phylogeny [24,25]. Therefore, in this study, we cloned the OAS genes of the emu, which is evolutionarily close to the ostrich, and investigated the structures and functions of emu OASs to elucidate their roles.

## 2. Materials and Methods

### 2.1. RNA Extraction

Emu spleen samples were kindly provided by Dr. K. Wada of the Tokyo University of Agriculture. Samples were collected after the emus were slaughtered for human consumption. Approval to collect the samples was obtained from the owner. Total RNA was extracted using TRIzol reagent (Thermo Fisher Scientific, Waltham, MA, USA) by crushing 200 mg of emu spleen, and the concentration of RNA was determined using SmartSpec™ (BioRad, Hercules, CA, USA). The extracted RNA was stored in a freezer at −80 °C until used.

### 2.2. Cloning

Reverse transcription reactions were performed using 1 μg of RNA to generate cDNA using ReverTra Ace^®^ (Toyobo Co., Ltd., Osaka, Japan). cDNA of *OAS*s was cloned using the 5′/3′-RACE Kit, 2nd generation (Roche Diagnostics GmbH, Mannheim, Germany), by following the manufacturer’s protocol. The primer sequences for PCR were based on the sequences predicted via whole-genome sequencing (XM_026117387.1, XP_009671383.1). Table 1 lists the primers used in this study. The emu *OAS* PCR products were cloned into a pGEM-T-Easy vector and transfected into DH5α, and the plasmids were extracted and sequenced. For functional analysis, PCR was performed using FLAG-tagged primers, and the PCR products were inserted into the pCAG-IRES-EGFP vector. Genetic recombination experiments were approved by Hokkaido University (approval number: 2020-034). After determining the OAS sequences, conserved motif analysis of emu OASs was performed using MOTIF Search (https://www.genome.jp/tools/motif/, accessed on 14 April 2024).

### 2.3. Phylogenetic Tree

Protein sequence alignment with other birds was performed using MEGA X software [26], and the reported protein sequences were: *Gallus gallus* (chOASL, BAB19016.1), *Anser cygnoides* (goOASL, ANW12075), *Anas platyrhynchos* (duOASL, ANW12076), and *Struthio camelus australis* (osOASL, XP_009671383, and osOAS1, XP_009667960). Phylogenetic trees were created using the neighbor-joining method. The amino acid sequences were aligned using a neighbor-joining algorithm, which created a tree based on the balanced minimum evolution criterion. The tree was drawn to scale with the branch lengths (next to the branches) in the same units as the evolutionary distances used to construct the phylogenetic tree.

### 2.4. Enzymatic Activity Assay

Enzymatic activity was measured as described previously [18,27]. The HEK293FT cell line was obtained from Dr. N. Sasaki of Kitasato University. HEK293FT cells were grown in Dulbecco’s modified Eagle’s medium (DMEM; Thermo Fisher Scientific) supplemented with 10% fetal bovine serum (Atlas Biological, Fort Collins, CO, USA) and 1% penicillin-streptomycin-L-glutamine solution (Fujifilm Wako Pure Chemical, Osaka, Japan) (final concentration: 100 unit/mL penicillin, 100 μg/mL streptomycin, 2 mM L-glutamine).

Enzymatic activity was determined by transfection of HEK293FT cells with a plasmid encoding OASs and lysis of the cells with Flag-lysis buffer (300 mM NaCl, 20 mM Tris-HCl at pH 7.4, 10% glycerol, 0.2% Triton X-100, and 5 mM β-mercaptoethanol) after 72 h. Then, 2.5 μL of lysate and 7.5 μL of reaction buffer (20 mM Tris-HCl at pH 7.4, 20 mM magnesium acetate, 2.5 mM dithiothreitol, 5 mM ATP, 50 μg/mL poly(I):(C), and 5 μCi of (α-^32^P) ATP (3000 Ci/mmol)) in a final volume of 10 μL were mixed and reacted at 37 °C for 24 h. Poly(I):(C) was used to activate OAS as an RNA mimic. Subsequently, the mixtures were incubated at 95 °C for 5 min to terminate the reaction, and electrophoresis was performed on a 20% urea acrylamide gel for 1 h. After electrophoresis, the gels were soaked in protection buffer (3% glycerol (*w*/*v*), 40% MeOH, 10% HOA) for 2 h and then dried for 1 h using a Model 583 Gel Dryer (BioRad). The dried gels were exposed to a BAS 2000 imaging plate (FUJIFILM, Tokyo, Japan) for 30 min and quantified using a BAS 2000 Image Analyzer (Fuji Film).

### 2.5. Antiviral Experiments

The inhibitory effect on flaviviral replication was measured using a previous method [28]. The BHK-21 cells were obtained from the American Type Culture Collection. Briefly, BHK-21 cells were grown in DMEM (Thermo Fisher Scientific) supplemented with 10% fetal bovine serum (Atlas biological) and 1% penicillin–streptomycin–L-glutamine solution (Fujifilm Wako Pure Chemical) (final concentration: 100 unit/mL penicillin, 100 μg/mL streptomycin, 2 mM L-glutamine). BHK-21 cells were seeded in 24-well plates at a density of 1.0 × 10^5^ cells/well. After 24 h, 10 μg of pIRES-EGFP (empty vector as a control), pemOAS1-EGFP, pemOASL-EGFP, posOAS1-EGFP, posOASL-EGFP, and pmOas1b-EGFP were transfected using Lipofectamine 2000^®^ (Thermo Fisher Scientific) according to the manufacturer’s protocol. posOAS1-EGFP, posOASL-EGFP, and pmOas1b-EGFP were constructed in a previous study, and mOas1b was cloned from MSM/Ms strain which is a wild-derived strain [18,28]. EGFP expression was observed using a fluorescence microscope to estimate the transfection efficiency. West Nile virus (WNV) replicon RNA, harboring the secreted alkaline phosphatase (SEAP) reporter gene instead of viral structural genes, was propagated by mMESSAGE mMACHINE^®^ Kit (Thermo Fisher Scientific), as previously reported [28]. Briefly, the WNV replicon DNA plasmid was linearized with Not I, and the single-stranded end was removed using mung bean nuclease. Then, 1 μg of linearized WNV replicon DNA was used as the template for transcription. WNV replicon RNA (500 ng) was lipofected into BHK-21 cells. The culture supernatants were collected after culturing the cells for 72 h post-lipofection and centrifuged under 12,000× *g* for 30 s. The supernatants were collected and stored at −80 °C until used. The amount of reporter protein in the culture supernatant was measured using Great EscAPe™ SEAP Chemiluminescence Kit 2.0 (Takara Bio Inc., Shiga, Japan) and an Infinite M200 PRO plate reader (TECAN Japan Co., Ltd., Kanagawa, Japan) according to the manufacturer’s protocol.

### 2.6. Statistical Analysis

The groups were compared with Dunnett’s test and data are shown as means ± standard error.

## 3. Results

### 3.1. Cloning

The purity of the extracted RNA was evaluated by a ratio of OD260/280. The values of ratio of OD260/280 were ranged in 1.9 ± 0.7. We cloned emu OASs to determine its amino acid sequence. Using the ostrich *OAS* sequence as a reference, we successfully determined the emu *OAS* sequence using RACE method. The accession numbers for emu *OAS1* and *OASL* were LC788476 and LC788477, respectively.

The amino acid sequences of the ostrich and emu OASs were compared (Figure 1A,B). Amino acid sequence homology was 80% for OAS1 and 78% for OASL. The sequences of the two species were compared in terms of regions I–III, which have been reported to be essential for the function of OASs; I and II sequences were completely identical, whereas the III sequence showed a little difference [18,19,29]. Domain structural analysis of these OASs showed that the OASL protein conserved three domains, the nucleotidyltransferase, OAS1_C, and two ubiquitin-like domains—UBL1 and UBL2—whereas osOAS1 possessed only the nucleotidyltransferase and OAS1_C domains.

### 3.2. Phylogenetic Tree

The emu OAS1 and OASL sequences were compared with those of other bird species. A phylogenetic tree was constructed using the neighborhood method (Figure 1C). The OASL sequence of the emu was similar to that of the ostrich and created a distinct group among the birds. Therefore, the emu and ostrich are unique bird species among birds due to possessing OAS1 with high homology.

### 3.3. Enzymatic Activity

The enzymatic activities of OASs to synthesize 2′-5′-oligoadenylates from ATP were measured. OAS enzymatic activity was assessed using lysates of HEK293FT cells transfected with the pIRES-EGFP empty vector as a control, pmOas1b-EGFP, posOASL-EGFP, posOAS1-EGFP, pemOASL-EGFP, and pemOAS1-EGFP. As previously reported [18,23], osOAS1 synthesized oligoadenylates, but osOASL and mOas1b did not have oligoadenylate synthesis activity (Figure 2). The results showed that emu OAS1 synthesized oligoadenylates via enzymatic activity, whereas no oligoadenylate synthesis was observed in emu OASL (Figure 2).

### 3.4. Inhibitory Activity on the WNV Replicon Replication

The replication inhibitory effects of OAS1 and OASL of the emu were compared with those of osOASs and mOas1b. Inhibition of WNV replicon replication was measured in BHK-21 cells transfected with the pIRES-EGFP empty vector as a control, pmOas1b-EGFP, posOASL-EGFP, posOAS1-EGFP, pemOASL-EGFP, or pemOAS1-EGFP (Figure 3A). The results showed that OASL of both emus and ostriches, as well as mOas1b, inhibited replication of the WNV replicon, whereas OAS1 of both emus and ostriches did not (Figure 3B).

## 4. Discussion

In this study, we succeeded in sequencing the OASs of the emu, which was evolutionarily close to the ostrich. The amino acid sequence homology was 80% for OAS1 and 78% for OASL compared to those of the ostrich. This was almost consistent with the results of previously reported homology comparisons of other genes between the two species [30,31]. The results of the phylogenetic tree analysis showed that the homologies of the OASLs could be compared and divided into Palaeognathae and Neognathae groups. Domain analysis revealed that OASL has three domains (nucleotidyltransferase, OAS1_C, and ubiquitin-like domain), similar to previously reported OASLs in other avian and mammalian species. These results suggest that the components of the OASL gene are conserved in a wide range of animal species and may be important for their function.

The anti-flavivirus replication activity of OASs was suggested by mOas1b as the gene responsible for the susceptibility to flavivirus infection in mice [32,33]. However, Oas1b does not have the enzymatic activity to synthesize oligoadenylates from ATP. This anti-flaviviral replication pathway was observed even when RNase L was knocked out, suggesting that there is a flaviviral replication inhibition pathway independent of the oligoadenylate synthetic activity [34]. The inhibition of flavivirus replication has also been reported in other avian species [18]. BHK cells are one of the most commonly used cells for viral infection experiments because they lack interferon induction [35,36]. The proteins that constitute the OAS/RNase L pathway are expressed at low levels in BHK cells because they are ISGs. Therefore, by measuring the replication of viral replicons in BHK cells, it is possible to measure the inhibitory activity of OAS on flavivirus replication by reducing the activation of the OAS/RNase L pathway through the enzymatic activity of OASs. In this experiment, the amount of reporter protein expressed by the WNV replicon was reduced by OASL, but not by OAS1, in both emus and ostriches. Alignment of the amino acid sequence of OAS1 showed that emu OAS1 had an additional 48 amino acids at the C-terminus compared to ostriches. This additional region may not affect anti-flaviviral activity. To elucidate the details of flavivirus replication inhibition by OASs, it is necessary to analyze common sequences and function-related domains by examining the sequences of OASs with flavivirus-specific viral replication inhibition in other animal species.

OASs synthesizes 2′-5′-oligoadenylates from ATP. Enzyme activity is activated by the recognition of double-stranded RNA. In this experiment, stimulation with poly(I):(C) resulted in the synthesis of oligoadenylates in OAS1 but not in OASL. This activity is believed to be the basic function of this enzyme, which is responsible for cellular viral defense [37]. Sequence comparison with the ostrich OAS1 revealed that the regions of the sequences that are thought to be important for enzymatic activity were conserved [29]. These regions were highly conserved in other avian species, which supports their functional importance. OASL has been reported to lack enzymatic activity in many mammalian species, indicating that it differs from chicken, geese, and ducks [14,16]. The OASL of ostriches and emus showed no oligoadenylate synthetic activity, suggesting that they have similar mammalian characteristics.

The division of the function of the OASs was found to develop in the emus as well as in the ostrich, although the significance of this division remains unclear. The OAS/RNase L pathway randomly cleaves and depletes intracellular single-stranded RNA [38]. Thus, it inhibits homeostatic protein synthesis in the cell. However, some ISGs evade degradation by RNase L to maintain their intracellular antiviral mechanisms. In mice, one ISG that escapes degradation by RNase L is mOas1b, which exhibits inhibitory effects on flaviviral replication without enzymatic activity. This protein also inhibits the oligoadenylate synthesis activity of mOas1a, which contributes to the OAS/RNase L pathway. In mice, it has been suggested that the division of functions and duplication of OASs allows rapid virus elimination with slightly suppressed OAS/RNase L by mOas1b during flaviviral infection, whereas the OAS/RNase L pathway by Oas1a is used for virus elimination during other viral infections [39]. Therefore, the division of functions may contribute to viral defense through an optimal mechanism depending on the virus species. The division of functions may be advantageous to host organisms, and the fact that emus and ostriches have two OAS genes, which play an allotted role in anti-flavivirus replication and enzymatic activity, may be advantageous in viral defense compared to other avian species that have only one OAS gene. The division of OASs functions into multiple OAS genes has also been observed in mammals (Table 2). In particular, mouse Oas1 paralogs have been well investigated for their anti-flaviviral replication and enzymatic activities [23]. In mice, only Oas1b has anti-flavivirus replication, Oas1a and Oas1g have enzymatic activity, and the other OAS1 paralogs do not have either activity. Although other mammals, such as humans, swine, and rats, have not been thoroughly examined for these activities, they have multiple OAS genes. The emu and ostrich were evolutionarily different from other birds in terms of OAS diversity. The enzymatically active OAS1 may be an ancestral OAS in invertebrates. Most birds, such as chicken, ducks, and geese, acquired anti-flavivirus activity from the same molecule, whereas the ratitae, such as the ostriches and emus, and mammals, such as the mice, acquired the same activity as the other OAS molecules, OASL and Oas1b. However, these evolutionary dynamics need to be clarified in future studies.

The results of this study suggest that emus might be more resistant to flaviviruses than other bird species. Experimental infections of WNV to birds has been conducted, and differences in susceptibility of bird species, including many wild birds, have already been reported [40]. Fatalities caused by WNV have been reported in a young ostrich, but not in adults [41]. However, there have been few reports of viral infections in emus, and the contribution of OAS to viral defense in emus remains unclear. Experimental viral infections are needed to deepen our knowledge of viral defense in emus. Hopefully, this study will open novel perspectives on defense against viruses in birds.

## 5. Conclusions

In this study, emu OAS were cloned to investigate their sequence and function and to elucidate their role in the emu. Cloning results showed that emus contained *OAS1* and *OASL* which were more closely related to those ostriches than to other birds. The amino acid sequence homology was 80% for OAS1 and 78% for OASL compared with ostriches. Functional investigations showed that emu OAS1 and emu OASL have enzymatic and anti-flaviviral activities, respectively, similar to those of ostriches.

The separation of OAS functions into multiple OAS genes, as seen in mammals, may be advantageous for the organism to defend against viral infection through an appropriate mechanism depending on the virus species. The emu and ostrich are evolutionarily different from most birds and may be more closely related to mammals with respect to OAS diversity.

## Figures and Tables

**Figure 1 animals-14-01579-f001:**
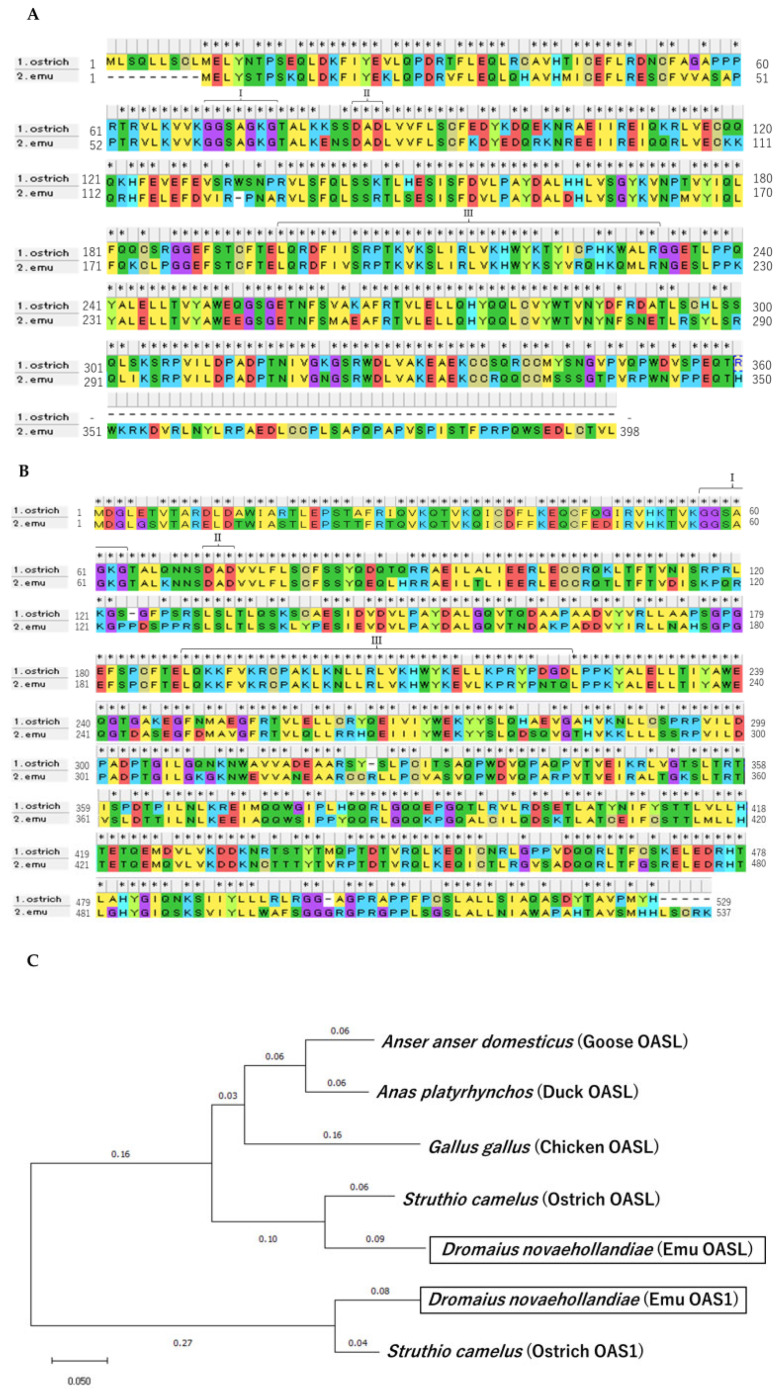
Cloning of the emu OAS sequence was performed to obtain the amino acid sequences of OAS1 (**A**) and OASL (**B**). These amino acid sequences were aligned with those of ostriches. Homologous amino acid sequences were marked with * at the top. I–III indicate regions essential for the OAS enzymatic function. (**C**) A phylogenetic tree of avian OASs with reported nucleotide sequences. The emu OASs of the sequences obtained in this study were put in squares.

**Figure 2 animals-14-01579-f002:**
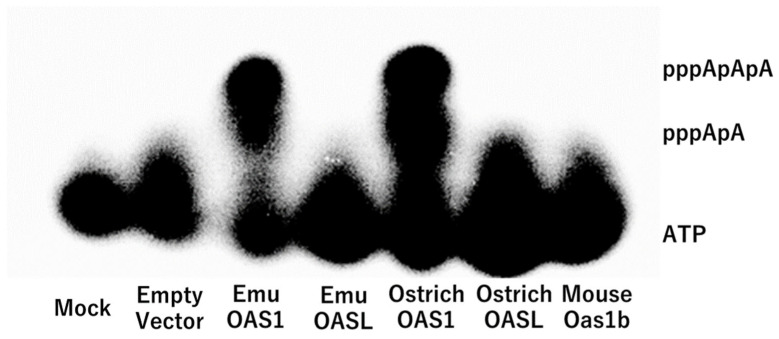
Oligoadenylate synthetase activity was determined for emu, ostrich, and mouse OASs. The products after the enzymatic reaction were electrophoresed and detected as oligomerized ATP with ^32^P.

**Figure 3 animals-14-01579-f003:**
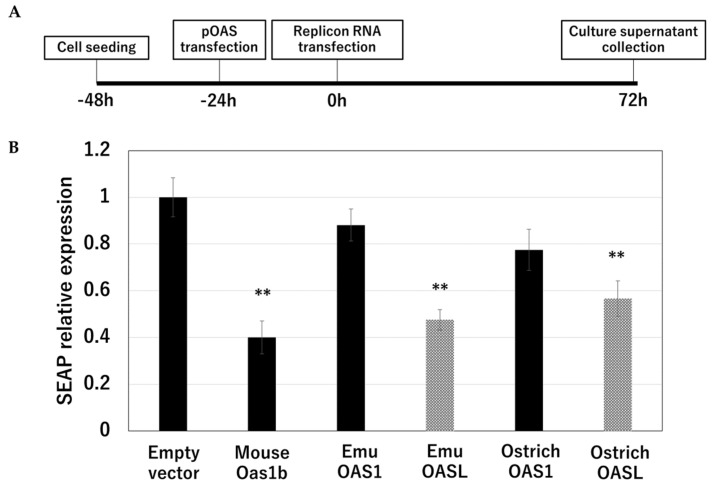
Inhibitory effect of emu OASs on WNV replication was measured. (**A**) Timeline of the antiflavivirus activity experiment. BHK-21 cells were seeded and then sequentially transfected with pOAS plasmid and WNV replicon RNA. The supernatant of the culture medium was collected after 72 h. (**B**) Inhibitory effect of emu OASs on WNV replication was measured. SEAP, a reporter protein in the culture supernatant, was measured. Error bars indicate standard errors. ** indicates *p* < 0.01 compared with empty vector.

**Table 1 animals-14-01579-t001:** Primers for 5′/3′ RACE and cloning of emu *OAS* cDNA.

Name	Direction	Primer Sequence
OAS1-5′RACE	Reverse	CCTCTGCTTCTTGCACTCCA
OAS1-3′RACE	Forward	CTGTCAGCACCTCAACCTGCA
OASL-5′RACE	Reverse	TACCAGTGCTTGACCAGGC
OASL-3′RACE	Forward	GCCTGGTCAAGCACTGGTA
OAS1-*Xba* Ⅰ	Forward	TGCTCTAGAGCAGCACGGGCGCTGTCACAG
OASL-*Xba* Ⅰ	Forward	TGCTCTAGAGTATGGATGGGCTGGAGA
Emu OAS1 FLAG-*Xba* Ⅰ	Reverse	TTATTATCTAGATCACTTGTCGTCATCGTCTTTGTAGTCGAGGACAGTGCAGAGGTC
Emu OASL FLAG-*Xba* Ⅰ	Reverse	TTATTATCTAGATCACTTGTCGTCATCGTCTTTGTAGTCGTTTATTTCCGGCATGATA

**Table 2 animals-14-01579-t002:** Diversity of OASs in birds and mammals.

Species	OAS Family	Reference
Birds		
Chicken	OASL *^,^**	[20]
Goose	OASL *^,^**	[18,19]
Duck	OASL *^,^**	[18]
Ostrich	OASL *, OAS1 **	[18]
Emu	OASL *, OAS1 **	This study
Mammals		
Human	OAS1 **, OAS2 **, OAS3 **, OASL1	[14]
Swine	OAS1a **, OAS1b **, OAS2, OASL	[15]
Rat	OAS1a, OAS1b, OAS1c, OAS1d, OAS1e, OAS1f, OAS1g, OAS1h, OAS1i, OAS2, OAS3, OASL	[12]
Mouse	Oas1a **, Oas1b *, Oas1c, Oas1d, Oas1e, Oas1f, Oas1g **, Oas1h, Oas2, Oas3, OasL1, OasL2	[23]

*, Anti-flaviral replication. **, enzymatic activity. The underlined OASs are reported not to have either of the two activities [20]. Other OASs not marked with asterisks are unknown for these two activities because of a lack of reports.

## Data Availability

The data that support the findings of this study are available from the corresponding author upon reasonable request.

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
