# Peer review of "Functional Analysis of Oligoadenylate Synthetase in the Emu (Dromaius novaehollandiae)"

_animals, 2024, doi:10.3390/ani14111579_

Round 1

Reviewer 1 Report

Comments and Suggestions for Authors

MANUSCRIPT ANIMALS-3016098 PEER REVIEW

The study focuses on the presence and functionality of Oligoadenylate synthetase (OAS), an essential antiviral protein found across various animal species, with a specific emphasis on the ostrich and its two genes, OAS1 and OASL. The ostrich possesses these two genes, with OAS1 exhibiting oligoadenylate synthetic activity, and OASL demonstrating inhibition of flavivirus-specific genome replication. However, the broader question of whether this gene duplication with divergent functions is exclusive to the ostrich remains unanswered. To address this, the researchers turned to the emu, a bird closely related to the ostrich. The study involved sequencing and cloning emu OAS genes and analyzing their functionalities. The findings revealed a significant amino acid sequence homology between emu and ostrich OAS genes, with emu OAS1 and OASL sharing approximately 80% and 78% homology, respectively, compared to their ostrich counterparts. Moreover, the amino acid sequences relevant to enzymatic function closely resembled those of ostriches. The functional analysis demonstrated that emu OAS1 exclusively exhibited OAS activity, while emu OASL solely inhibited flaviviral replication. These results underscored the similarity between emus and ostriches concerning OAS genes, suggesting a shared evolutionary trajectory in terms of antiviral defense mechanisms among these birds. Overall, the study sheds light on the evolutionary dynamics of viral defense mechanisms in avian species, specifically highlighting the conservation and functional specialization of OAS genes in the ostrich and its close relative, the emu.

The manuscript seems well written, with questions appropriately addressed by sound methods and unambiguous results. Before acceptance for publication, I would like the authors to consider some minor revisions, as described below:

ABSTRACT

Lines 19-20. The authors claim that the study “… provides insights into the evolution of viral defense mechanisms in birds”. How so? May be there is not like too much room for explanations in this section, but there is no elaboration in the rest of the manuscript whatsoever. I suggest the authors to better explain and discuss such claims where appropriate.

INTRODUCTION

MATERIALS AND METHODS

Lines 108-114. The Phylogenetic tree reconstruction explanation is rather poor in detail. I suggest the authors to elaborate on how phylogenetic tree reconstruction was made.

Minor and typos

MATERIALS AND METHODS

Line 112. Neighborhood seems misspelled.

DISCUSSION

Line 275. It says “thesequences”. Must say “the sequences”.

Author Response

We appreciate your insightful comments, which have helped us in significantly improving the manuscript.

ABSTRACT

Lines 19-20. The authors claim that the study “… provides insights into the evolution of viral defense mechanisms in birds”. How so? May be there is not like too much room for explanations in this section, but there is no elaboration in the rest of the manuscript whatsoever. I suggest the authors to better explain and discuss such claims where appropriate.

Answer: It is considered that birds have a common ancestor and were separated into Palaeognathae and Neognathae at first (https://www.mdpi.com/2673-6004/2/1/1). As stated in Discussion, this study suggests that the features of OASs are conserved in Palaeognathae and that they have a different evolutionary direction for viral defense. Therefore, we decided that the suitable word would be “Palaeognathae” rather than “bird” in page1, Line19. Also, we replace “viral defense mechanism” into “viral defense by OAS protein family” since we investigated only OAS family in this study.

MATERIALS AND METHODS

Lines 108-114. The Phylogenetic tree reconstruction explanation is rather poor in detail. I suggest the authors to elaborate on how phylogenetic tree reconstruction was made.

Answer: We added sentences of details of the neighbor-joining method to create Phylogenetic tree.

Minor and typos

MATERIALS AND METHODS

Line 112. Neighborhood seems misspelled.

DISCUSSION

Line 275. It says “thesequences”. Must say “the sequences”.

Answer: Thank you for pointing out.

  • We have corrected “Neighbor-joining” on Line 114 of the revised manuscript.
  • We have revised on Line 270 of the revised manuscript.

Reviewer 2 Report

Comments and Suggestions for Authors

Dear Authors of the Manuscript entitled “Functional analysis of oligoadenylate synthetase in the emu 2 (Dromaius novaehollandiae)”, I have thoroughly read the Manuscript and I found it scientifically important, methods well used and writing correct. The Manuscript provides information on Oligoadenylate synthetase genes in emu having in mind the fact that ostrich, which is the only species that has two genes OAS1 and OASL is a close relative. The results showed that emu OAS1 has OAS activity and OASL only inhibited flaviviral replication. Before accepting this manuscript there are several issues listed in the text below that I would like to see improved:

Line 67-72. Comment: This paragraph seems to be better fitting in the text before since it gives information on OAS family. Kindly consider to move this paragraph before.

Line 86-90. Please give uniform company names including city and state as in text afterwards.

Line 74 and throughout the text. Please give Latin name of species when they appear for the first time in the text.

Line 86-88 Was there an owner s permission for sampling at the slaughterhouse or under which contract the samples were taken? Please specify.

Line 151. Were BHK cell used in the experiment also maintained with 10% FBS? At this rate of FBS in 72% this cell line usually deteriorates. Please specify those experimental conditions.

Results. Please give the results of RNA quality check. In material it is said that the RNAs were checked but the outcome is not stated.

Line 199-202. This sentence belongs more to material and methods section.

Line 215-219. This sentence belongs more to material and methods section.

Line 224-231. This paragraph contains repetitions from introduction.

Line 315-316. This is a strong statement based on in vitro experimental results. I suggest using “emus might be more resistant”.

Author Response

We appreciate your insightful comments, which have helped us in significantly improving the manuscript.

Line 67-72. Comment: This paragraph seems to be better fitting in the text before since it gives information on OAS family. Kindly consider to move this paragraph before.

Answer: As Reviewer 2 had mentioned, we changed the position of the paragraph.

Line 86-90. Please give uniform company names including city and state as in text afterwards.

Answer: We have modified both city and state to be listed in the company name for the first time and only in the company name for the second and subsequent times in Materials and Method section.

Line 74 and throughout the text. Please give Latin name of species when they appear for the first time in the text.

Answer: According to reviewer 2 comment, we added the Latin name.

Line 86-88 Was there an owner s permission for sampling at the slaughterhouse or under which contract the samples were taken? Please specify.

Answer: We obtained permission from the owner for the sampling. We added a description about the permission in Line 89 of revised manuscription.

Line 151. Were BHK cell used in the experiment also maintained with 10% FBS? At this rate of FBS in 72% this cell line usually deteriorates. Please specify those experimental conditions.

Answer: We treated BHK-21 cell with handling information by American Type Culture Collection (ATCC) [1]. Throughout this study, we cultured BHK-21 cell in DMEM with 10% FBS without any problem. Considering the damages to cells caused by lipofection and WNV replicon, we decided the density of seeded cells in the antiviral experiments.

[1] BHK-21 [C-13] https://www.atcc.org/products/ccl-10

Results. Please give the results of RNA quality check. In material it is said that the RNAs were checked but the outcome is not stated.

Answer: According to reviewer 2 comment, we added the sentence of RNA quality check by the ratio of OD260/280 in Line 183-184 at the first part of results.

Line 199-202. This sentence belongs more to material and methods section.

Line 215-219. This sentence belongs more to material and methods section.

Line 224-231. This paragraph contains repetitions from introduction.

Answer: As Reviewer 2 had mentioned, overlapping parts were deleted or moved to the correct section.

Line 315-316. This is a strong statement based on in vitro experimental results. I suggest using “emus might be more resistant”.

Answer: As Reviewer 2 had mentioned, we changed the sentence to more suitable expression “emus might be more resistant” in Line 310-311 of revised manuscription.

Round 2

Reviewer 2 Report

Comments and Suggestions for Authors

Dear Authors,

thank you for your reply. I find the changes sufficient and that they improved the quality of the Manuscript which is now acceptable for publication.